Corrected: Publisher correction

# Mapping microscale wetting variations on biological and synthetic water-repellent surfaces

Ville Liimatainen [1], Maja Vuckovac[2], Ville Jokinen[3], Veikko Sariola[1,4], Matti J. Hokkanen[1,2], Quan Zhou[1] & Robin H.A. Ras [2,5]

Droplets slip and bounce on superhydrophobic surfaces, enabling remarkable functions in biology and technology. These surfaces often contain microscopic irregularities in surface texture and chemical composition, which may affect or even govern macroscopic wetting phenomena. However, effective ways to quantify and map microscopic variations of wettability are still missing, because existing contact angle and force-based methods lack sensitivity and spatial resolution. Here, we introduce wetting maps that visualize local variations in wetting through droplet adhesion forces, which correlate with wettability. We develop scanning droplet adhesion microscopy, a technique to obtain wetting maps with spatial resolution down to 10 μm and three orders of magnitude better force sensitivity than current tensiometers. The microscope allows characterization of challenging non-flat surfaces, like the butterfly wing, previously difficult to characterize by contact angle method due to obscured view. Furthermore, the technique reveals wetting heterogeneity of micropillared model surfaces previously assumed to be uniform.

[1] Department of Electrical Engineering and Automation, Aalto University School of Electrical Engineering, Maarintie 8, 02150 Espoo, Finland. [2] Department of Applied Physics, Aalto University School of Science, Puumiehenkuja 2, 02150 Espoo, Finland. [3] Department of Chemistry and Materials Science, Aalto University School of Chemical Engineering, Tietotie 3, 02150 Espoo, Finland. [4] Faculty of Biomedical Sciences and Engineering, Tampere University of Technology, Korkeakoulunkatu 3, 33720 Tampere, Finland. [5] Department of Bioproducts and Biosystems, Aalto University School of Chemical Engineering, Kemistintie 1, 02150 Espoo, Finland. Ville Liimatainen and Maja Vuckovac contributed equally to this work. Correspondence and requests for materials should be addressed to Q.Z. (email: quan.zhou@aalto.fi) or to R.H.A.R. (email: robin.ras@aalto.fi)

Superhydrophobic surfaces enable exceptional functions in biology and technology[1–5]. Understanding how water-repellency emerges from the microscale and nanoscale features[1] is critical to advance the development of these surfaces[6–8]. Biological superhydrophobic surfaces often contain irregular surface texture and details[9, 10], such as creases or veins, and synthetic surfaces are prone to fabrication defects. Such irregularities in surface texture and chemical composition lead to spot-to-spot variation of wetting properties[11, 12], which may affect or even govern droplet mobility[13, 14], icing[15], and condensation[16]. Even though these variations have been considered in theory[1], so far they have not been probed experimentally, partly because existing contact angle and force-based methods lack sensitivity and spatial resolution[17–19]. The contact angle method, describing a surface by a single pair of apparent advancing and receding contact angle values, is still viewed as the gold standard in hydrophobic surface characterization. As the measurement is based on observing a moving contact line, it is inherently unsuitable for precise spatial mapping. Moreover, as an optical method, contact angle measurements become increasingly inaccurate for contact angles beyond 150°[17, 18] due to resolution limit of the optical system, and often suffer from obscured view of the contact line on curvy surfaces[9]. Wetting properties have also been characterized by droplet friction forces, i.e., resistance to lateral motion[20–22] and by droplet adhesion forces, i.e., resistance to detaching a droplet in the normal direction[8, 19, 23, 24]. It has been experimentally verified that snap-in (first droplet contact) and pull-off (droplet separation) adhesion forces on hydrophobic surfaces are related to, respectively, the advancing contact angle ($\theta_{adv}$) and the receding contact angle ($\theta_{rec}$)[19]. Smaller forces correspond to larger contact angles. However, current tensiometers are not sensitive enough to detect droplet adhesion forces below 1 µN. On the other hand, scanning probe microscopy techniques, e.g., atomic force microscopy (AFM), can have nN or even better force resolution, though so far have not been used to map spatial wetting variations. To study wetting in ambient air, the water droplet must be large enough so that evaporation during a single measurement becomes insignificant, and nN sensitive AFM cantilevers cannot support such large droplets in air. Droplet–surface interactions have therefore previously been measured by AFM not in ambient air but instead in liquid–liquid and bubble-liquid systems[25], or by using the AFM tip itself as the surface under study[24], hence greatly limiting the scope of AFM to study wetting of surfaces.

Here, we introduce scanning droplet adhesion microscopy, a technique for obtaining wetting maps of surfaces based on high-precision force measurements. The technique enables us to map in extraordinary detail the wetting forces, even on challenging, non-flat, biological surfaces. Since butterfly wings and their eyespots have sparked tremendous scientific interest[26, 27], we select

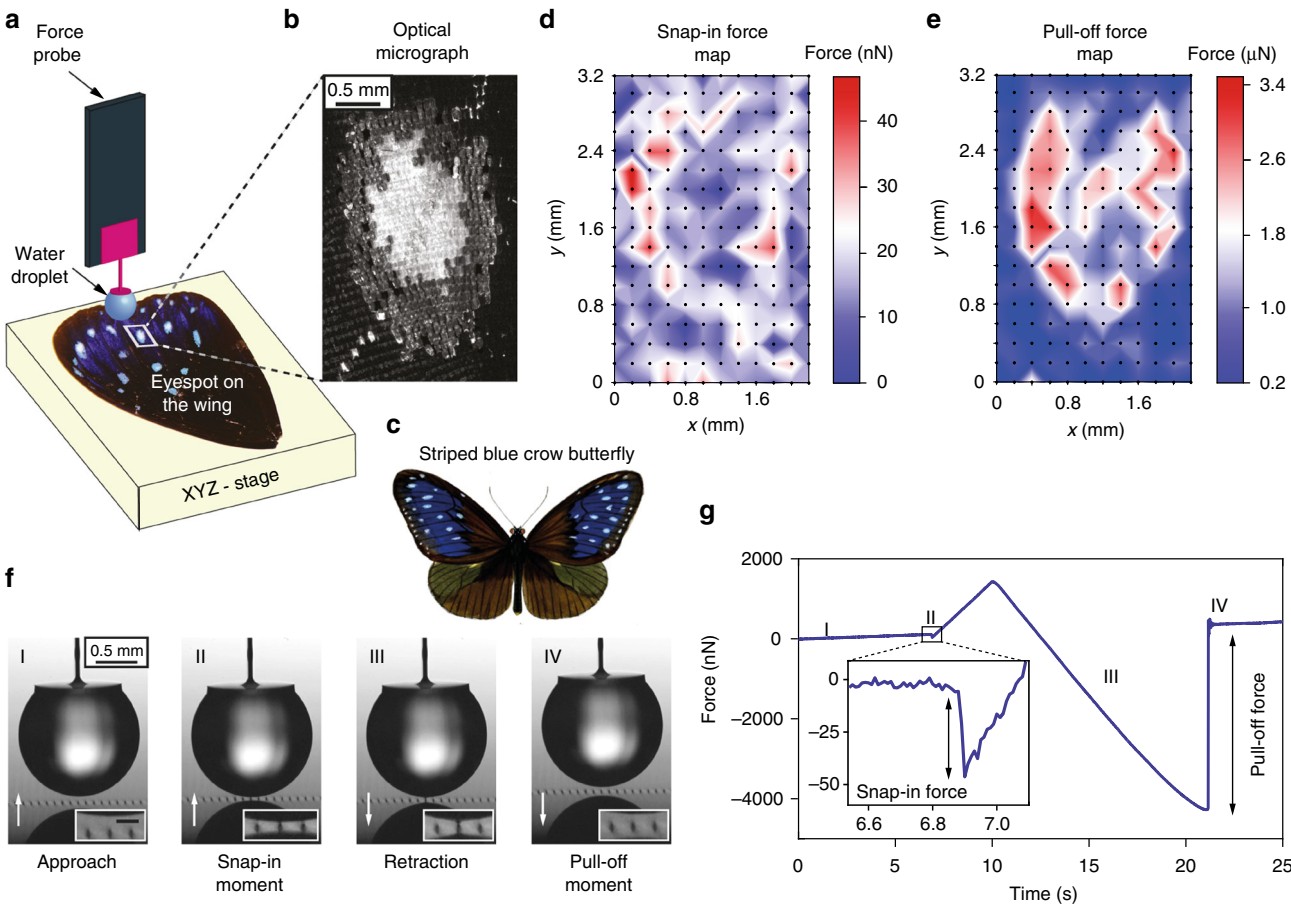

**Fig. 1** Concept of scanning droplet adhesion microscopy to construct wetting maps. **a** Schematic diagram of the microscope (not to scale). **b** Optical micrograph of scanned eyespot area on the wing of **c**, striped blue crow butterfly (typical wing span of adult specimen 80–90 mm; image by Frederic Moore, PD-1923); with corresponding **d** snap-in and **e** pull-off force maps. Dots on **d**, **e** denote the measurement points with 200 µm spacing. Colours in the maps denote measured force values, linearly interpolated between the data points. **f** Snapshots of individual measurement on a single hydrophobic 5 µm radius pillar (white arrows indicate direction of sample surface movement, inset scale bar 70 µm), and **g** corresponding typical force curve. Roman numerals in **f**, **g** indicate corresponding moments in the measurement

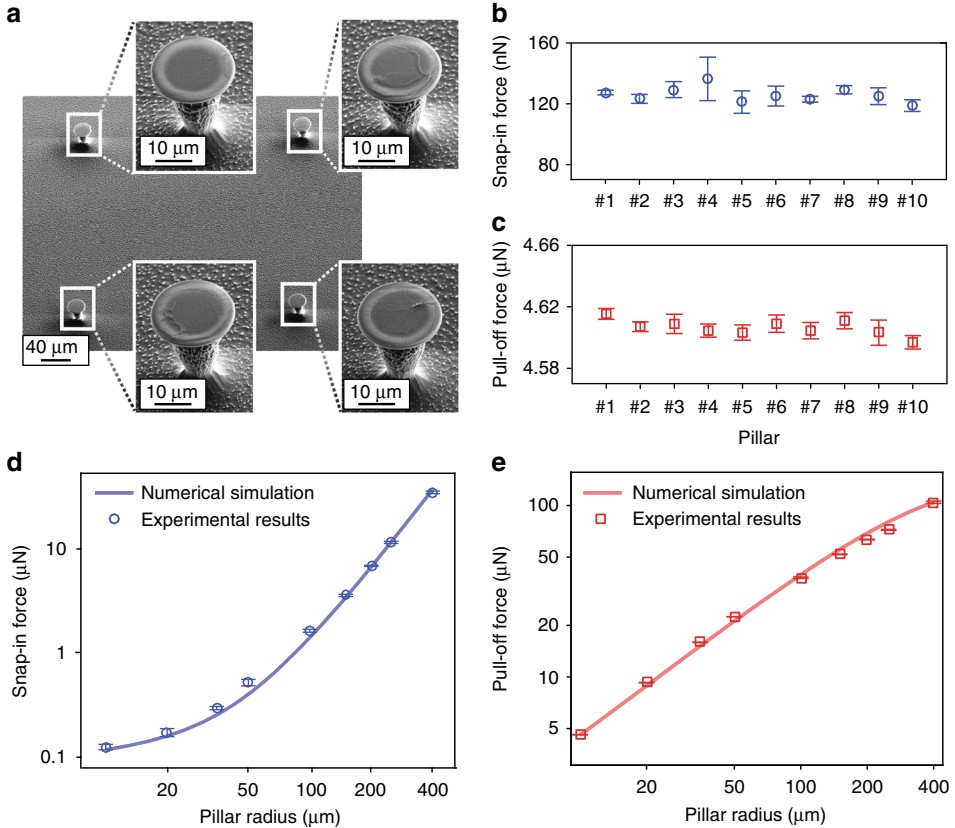

**Fig. 2** Accuracy and repeatability of droplet adhesion force measurements. Silicon micropillars with undercut tops were used for the accuracy and repeatability measurements. **a** SEM micrographs of four different 10 μm radius pillars showing tiny structural variations on the pillar tops. **b**, **c** Snap-in and pull-off force data from ten different 10 μm radius pillars. Data for 20–50 μm radius pillars are in Supplementary Fig. 2. **d**, **e** Snap-in and pull-off forces as function of pillar radius, the number of data points was 100 (10–50 μm radius pillars) or three (100–400 μm radius pillars). Solid lines show numerical simulation results. Error bars denote standard deviations in force

the wing of the striped blue crow butterfly (*Euploea mulciber*) as a model surface. Wetting maps obtained on an eyespot of the wing show correlation between wetting and visual appearance. We also show that on micropillared model surfaces, the wetting forces can vary pillar-to-pillar.

## Results

**Measurement concept**. The apparatus we built comprises a vertically mounted force sensor with a liquid droplet probe (e.g., water) and a multi-axis sample stage (Fig. 1a and Supplementary Fig. 1) for measuring normal forces point-by-point in a fully automated manner (Supplementary Movie 1). By scanning an eyespot area (Fig. 1b) on the striped blue crow butterfly wing (Fig. 1c), we obtain snap-in (Fig. 1d) and pull-off (Fig. 1e) force maps. Force measurement on a single point starts with moving up the sample surface to approach the droplet, followed by first contact (snap-in) with the droplet, then the sample surface moves down until it separates from the droplet (pull-off) (Fig. 1f). A typical force curve with nanonewton resolution, recorded at 100 Hz sampling rate, is shown in Fig. 1g. The droplet volume is refilled to 1.5 μl before every measurement; the effect of evaporation is not significant as only ~1% of the volume evaporates during a typical single measurement lasting around 20 s.

**Accuracy and repeatability**. To verify accuracy and repeatability of the microscope, we measured snap-in and pull-off forces on single silicon pillars (Fig. 2a–c and Supplementary Fig. 2) of varying radius, one pillar at a time. For each radius, the measurement is carried out on ten different pillars, with ten

repetitions on each pillar. The mean snap-in force for ten 10 μm radius pillars is 125.9 nN with pooled standard deviation of 6.6 nN (Fig. 2b), and the mean pull-off force is 4606.4 nN with pooled standard deviation of 5.3 nN (Fig. 2c). Analysis of variance indicates that pillar-to-pillar variation of snap-in force is six times larger than within-pillar variation and nine times larger in case of pull-off force. For the larger pillars, this difference is even clearer (Supplementary Table 1). We are thus capable of measuring tiny but significant differences in adhesion force from one pillar to another, likely caused by tiny structural variations on the pillar tops (see Fig. 2a). The 6.6 nN standard deviation in within-pillar measurements is predominantly attributed to sensor noise of the microscope (measured as 5 nN), verifying excellent repeatability of the method. We measured snap-in and pull-off forces for pillar radii between 10 and 400 μm. The results (Fig. 2d, e) show high consistency for each radius and agree well with numerical simulation. The simulation uses Young–Laplace equation and boundary conditions to solve the shape of the droplet, which can be used to compute the forces[18]. A detailed discussion on modelling and simulation can be found in Supplementary Note 1.

**Wetting forces on superhydrophobic surfaces**. We also compared snap-in and pull-off forces to optically measured contact angles for various hydrophobic and superhydrophobic surfaces. The results are shown in Supplementary Table 2 and Supplementary Figs. 3 and 4. The smallest measured snap-in force, 7.8 ± 1.9 nN, was obtained on Hydrobead-coated silicon wafer with optically measured contact angles $\theta_{adv}/\theta_{rec} = 169°/168°$. The smallest pull-off force, 218.5 ± 9.7 nN, was measured on a silicon

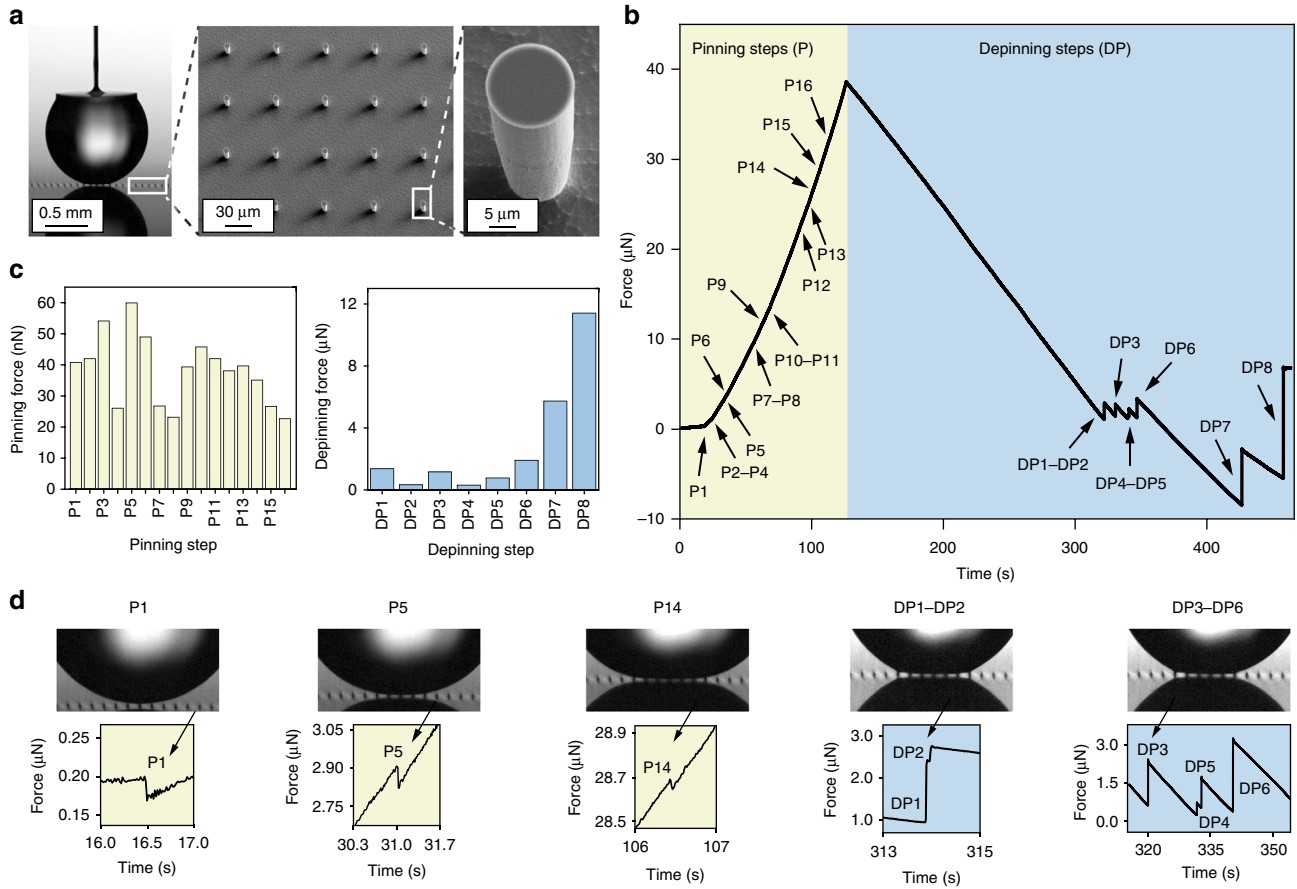

**Fig. 3** Multiple wetting steps during droplet advancing and receding. **a** Optical micrograph with SEM insets of the pillar surface (5 μm pillar radius, 70 μm spacing). **b** While a water droplet advances (sample moving up) 16 consecutive pinning steps (P1–P16) were detected, and 8 depinning steps (DP1–DP8) while droplet recedes (droplet evaporating, sample not moving). **c** Force values of individual pinning/depinning steps. **d** Selected details of droplet adhesion force curve and corresponding side view of droplet contact area. Details of all the steps are shown in Supplementary Fig. 5 and in Supplementary Movie 2

nanograss surface coated with fluoropolymer, where the optically measured contact angles were $\theta_{adv}/\theta_{rec} = 175°/169°$. In contrast to a recent study[19], there seems to be no contact angle limit beyond which the snap-in or pull-off force becomes zero. Therefore, using force sensors with enhanced sensitivity should allow characterization of even higher contact angle superhydrophobic surfaces.

Micropillar arrays are common model superhydrophobic substrates[1–3, 28, 29], and water droplets have been imaged to advance and recede step-wise, respectively, by pinning and depinning from pillar to pillar[29, 30]. Droplets rolling off such micropillar surfaces may oscillate due to the pinning and depinning steps[31, 32]. Here we quantify, for the first time, the tiny forces during pinning and depinning in great detail (Fig. 3). While a water droplet advances, in total 16 pinning steps were detected with forces ranging from 20 to 60 nN. Eight depinning steps were detected during droplet receding, with forces ranging from hundreds of nN to more than 10 μN. The butterfly wing exhibits similar pinning/depinning steps (Supplementary Fig. 6). The number of pinning steps in Fig. 3 is much higher than the number of depinning steps, because depinning from multiple pillars are combined into larger steps. It should be noted that the magnitude of pinning steps shown in Fig. 3c corresponds to snap-in force on an individual pillar, shown in Fig. 1g. On the other hand, the dynamics of collective depinning involves lateral deformation of the meniscus, so the magnitude of depinning steps in normal direction (Fig. 3c) is smaller than pull-off on individual pillars (Fig. 1g), except for the final depinning step. It should also

be noted that during a long measurement such as the one presented in Fig. 3, the volume loss due to evaporation of the droplet may affect the magnitude of the pinning and depinning steps (see Supplementary Note 1 for further theoretical discussion on the volume dependency).

**Wetting force mapping.** To challenge the sensitivity of the scanning droplet adhesion microscope, we measured snap-in and pull-off forces on a superhydrophobic-superhydrophobic patterned surface consisting of parallel stripes ($\theta_{adv}/\theta_{rec} = 175°/174°$) on background ($\theta_{adv}/\theta_{rec} = 173°/170°$) (Fig. 4a, b). $\theta_{adv}$ and $\theta_{rec}$ are all beyond 170°, and thus difficult to determine accurately with contact angle measurement[18]. We performed a line scan across the stripes, recording snap-in and pull-off forces every 75 μm. Whereas the snap-in force is too small to be detected for this surface, the pull-off force shows a clear difference between the stripes and the background (Fig. 4c).

While contact angle measurements of biological surfaces are often impossible due to an obscured baseline (Supplementary Fig. 7), we constructed detailed wetting force maps of striped blue crow butterfly wing. We probed an area of 2.2 × 3.2 mm covering one of the eyespots on the wing, with 200 μm spacing between measurement points, resulting in snap-in and pull-off force maps (Fig. 1b, d, e). We probed furthermore a smaller area (400 × 50 μm) of the eyespot in greater detail, with 10 μm spacing between measurement points (Fig. 4d, e). The corresponding snap-in and pull-off force maps are shown in Fig. 4f, together with an optical micrograph of the area. The wetting force maps reveal variations

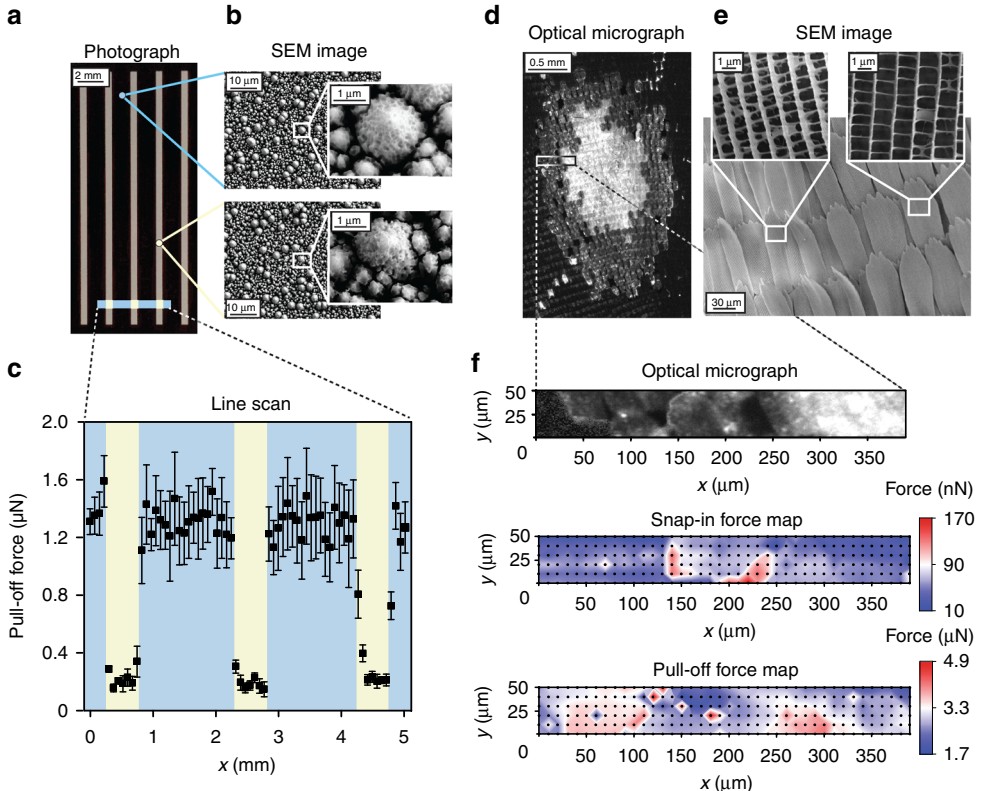

**Fig. 4** Detecting spatial variation of small (nN to μN) wetting forces. **a–c** Line scan at 75 μm resolution over superhydrophobic-superhydrophobic patterns of silicon nanograss with slightly different surface topography shows consistent difference in pull-off force. The error bars in **c** denote standard deviation of three measurements. **d–f** Area scan at 10 μm resolution over striped blue crow butterfly wing eyespot, where optically and topographically different surface structure is reflected in the corresponding wetting force maps

in snap-in and pull-off force when moving across the edge of the eyespot. The wetting variations correspond to variations in structural colour, and are thus due to subtle structural differences of the wing surface (scanning electron microscopy (SEM) micrographs of the probed area in Fig. 4e).

## Discussion

Microscale spatial heterogeneity is generally acknowledged as a major source of contact angle hysteresis and droplet friction[1], though how wetting on model surfaces and especially on irregular surfaces is related to the microscopic surface features is still not completely understood. Scanning droplet adhesion microscopy provides the sensitivity and the resolution that is critical to create wetting maps, a concept for hydrophobic surface characterization to study this relation in both biology and materials science. Using a sensor probe with increased sensitivity in pN range, measurement could become possible of superhydrophobic surfaces exhibiting even lower adhesion forces. Combination of scanning droplet adhesion microscopy with optical microscopy techniques would open the way to multimodal characterization of pinning and depinning events towards comprehensive understanding of wetting. Whereas this study focused on water, arguably the most relevant probe liquid, the scanning droplet adhesion microscopy could in principle be extended to study wetting of other liquids, including organic liquids on oleophobic surfaces.

## Methods

**Silicon micropillars**. Silicon micropillars were fabricated by deep reactive ion etching. First, a plasma-enhanced chemical vapour deposited (PECVD) oxide (Oxford PlasmaLab 80+, 300 °C, 8.5 sccm $SiH_4$, 1000 mTorr, 20 W) was deposited on a 4-inch silicon wafer (<100>, p-type doping 1–20 ohm-cm). The thickness of the oxide was 750 nm (12 min deposition time). The oxide was patterned by UV

lithography and reactive ion etching (Oxford PlasmaLab 80+, 25 sccm Ar, 25 sccm $CHF_3$, 200 W, 30 mTorr, 21 min etching time). The pillar radius was 5 μm. The silicon pillars were then etched by cryogenic deep reactive ion etching (Oxford PlasmaLab System 100, −110 °C, 40 sccm $SF_6$, 6 sccm $O_2$, 1050 W ICP power, 3 W platen power, 8 mTorr). The etch depth was 20 μm. The oxide mask was then removed by hydrofluoric acid. Finally, a thin hydrophobic fluoropolymer coating was deposited on top of the pillars by PECVD (Oxford PlasmaLab 80+, 100 sccm $CHF_3$, 50 W, 30 mTorr, 5 min).

**Silicon undercut micropillars**. Silicon undercut micropillars with a silicon dioxide top were fabricated for the accuracy and repeatability experiments. Undercut pillars with radius 10 μm - 50 μm were fabricated with the following process: a thermally oxidized (oxide thickness 1.2 μm) silicon wafer was used as a substrate. The oxide was patterned by UV lithography and reactive ion etching (Oxford PlasmaLab 80+, 25 sccm Ar, 25 sccm $CHF_3$, 200 W, 30 mTorr, 38 min). The samples were then etched anisotropically by cryogenic deep reactive ion etching (Oxford PlasmaLab System 100, −110 °C, 40 sccm $SF_6$, 6 sccm $O_2$, 1050 W ICP power, 3 W platen power, 8 mTorr, 10 min) to the depth of 22 μm. Finally, an isotropic silicon etching step was performed to create the undercut (Oxford PlasmaLab 80+, 100 sccm $SF_6$, 100 W, 100 mTorr, 18 min). Undercut pillars with radius 75 μm - 200 μm were fabricated by a process reported before[33].

**Silicon nanograss coated with fluoropolymer**. The substrate was a 4-inch silicon wafer (<100>, p-type doping 1–20 ohm-cm). Silicon nanograss was created by maskless cryogenic deep reactive ion etching (the so-called black silicon process) (Oxford PlasmaLab System 100, −110 °C, 40 sccm $SF_6$, 18 sccm $O_2$, 1000 W ICP power, 6 W platen power, 10 mTorr, 7 min etching time). The nanograss was made superhydrophobic by a PECVD deposition of a fluoropolymer thin film (Oxford PlasmaLab 80+, 100 sccm $CHF_3$, 50 W, 30 mTorr, 5 min deposition time).

**Fluoropolymer on Si wafer**. Fluoropolymer coated silicon surface was fabricated by the same PECVD process as with the fluoropolymer coated silicon nanograss. The substrate was an unprocessed 4 inch silicon wafer (<100>, p-type doping 1–20 ohm-cm).

**Silicon nanograss patterned surface for line scan**. To pattern the nanograss, a 500 nm thick silicon dioxide film was deposited (Oxford PlasmaLab 80+, 300 °C,

8.5 sccm SiH$_4$, 1000 mTorr, 20 W, 8 min) as a hard mask. The oxide mask was patterned by UV lithography and reactive ion etching (Oxford PlasmaLab 80+, 25 sccm Ar, 25 sccm CHF$_3$, 200 W, 30 mTorr, 14 min). To modify the nanograss topography, a silicon etching step (Oxford PlasmaLab 80+, 100 sccm SF$_6$, 100 W, 100 mTorr, 3 min) was performed. After this, the oxide mask was stripped in hydrofluoric acid and a second 3-min plasma etching step was performed with the same recipe (without a mask). Finally, the patterned nanograss was coated with the same PECVD fluoropolymer as the plain nanograss (previous section).

**Glaco on Si wafer**. Test-grade silicon (100) wafers were cleaned with an alkaline solvent (Deconex 11 Universal, VWR), rinsed thoroughly with Milli-Q water, and dried under nitrogen flow. Commercial Glaco Mirror Coat Zero (Soft99 Co.) was coated on the wafer by spraying followed by drying for 1 h.

**Hydrobead on Si wafer**. Test-grade silicon (100) wafers were cleaned with an alkaline solvent (Deconex 11 Universal, VWR), rinsed thoroughly with Milli-Q water, and dried under nitrogen flow. Afterwards, commercial Hydrobead Standard was applied on Si wafer and dried for 10 min.

**Silicone nanofilaments on Si wafer**. Surface was prepared as described in the literature[34]. Silicon (100) wafer was cleaned by ultrasonication in alkaline solvent (Deconex 11 Universal, VWR), rinsed thoroughly by Milli-Q water, and dried under nitrogen flow. Synthesis of the nanofilaments was done by chemical vapour deposition in an in-house built gas-phase reactor operating at atmospheric pressure. The reactor was purged with dry argon followed by pre-humidified argon until the relative humidity reached ca. 30% followed by sealing the reactor and injecting methyltrichlorosilane (99%, Aldrich) through a silicone septum. Methyltrichlorosilane evaporated and nanofilaments were grown onto the surface.

**Copper (II) hydroxide nanowires**. Copper (II) hydroxide nanowires were prepared as described in literature[35]. (NH$_4$)$_2$S$_2$O$_8$ (98%, Aldrich), NaOH (Fluka), HCl (Aldrich), and lauric acid (Fluka) were used as obtained. Mechanically polished copper substrate was immersed in HCl aqueous solution for 1 min, followed by washing with Milli-Q water and drying with nitrogen flow. Subsequently, the substrate was immersed into a mixed solution of 2.5 M NaOH and 0.13 M (NH$_4$)$_2$S$_2$O$_8$ for 20 min at room temperature and washed afterwards with Milli-Q water and dried under nitrogen flow. Chemical modification was performed by immersing substrate into 5 mM lauric acid (ethanol solution) for 20 min, followed by rinsing with Milli-Q water and drying under ambient conditions for a few minutes.

**PDMS replica of silicon nanograss**. Polydimethylsiloxane (PDMS) replica of silicon nanograss was fabricated as described in literature[36]. Cryogenic deep reactive ion etching was used to fabricate the silicon nanograss master on a 100 mm silicon wafer. PDMS mixture with base-to-curing ratio of 10:1 was prepared and poured over the silicon nanograss master, and then baked in an oven at 75 °C for 90 min. The cured PDMS was peeled off the master and functionalized with (1H,-1H,-2H,-2H-perfluorooctyl)-trichlorosilane inside a vacuum desiccator at room temperature. The silanization time was 12 h.

**Silicon nanograss coated with parylene**. Parylene-C coated silicon nanograss was prepared on a 4-inch silicon <100> wafer. Silicon nanograss was created by cryogenic deep reactive ion etching (Oxford Plasmalab System 100, −110 °C, 40 sccm SF$_6$, 18 sccm O$_2$, 1000 W ICP power, 6 W platen power, 10 m Torr, 7 min etching time). In the next step, approximately 250 nm of parylene-C was deposited on the silicon nanograss using a parylene deposition system (SCS Labcoter 2 PDS 2010). Finally, the surface was fluorinated in a reactive ion etcher (Oxford PlasmaLab 80+, 100 sccm SF$_6$, 100 W, 20 mTorr, 1 min), resulting in 200 nm parylene-C thickness.

**Fluoroalkyl self-assembled monolayer on Si wafer**. A silicon wafer was cleaned and activated with oxygen plasma treatment (PVA Tepla, 1000 W, 500 ml min$^{-1}$ O$_2$, 1 min). The sample was then put on a hotplate at 75 °C in a closed container together with 0.5 ml of (1H,-1H,-2H,-2H-perfluorooctyl)-trichlorosilane. The silanization time was 2.5 h.

**Butterfly wings**. Golden birdwing (*Troides aeacus*) and striped blue crow (*Euploea mulciber*) butterfly wings were purchased from Beijing Jiaying Store of Arts and Insects on taobao.com.

**Measurement setup**. The measurement setup (Supplementary Fig. 1) consists of a microforce sensing probe (FT-S100, range ± 100 μN and FT-S1000, range ± 1000 μN; FemtoTools AG, Switzerland) and a data acquisition board (NI USB-6351, National Instruments Inc., USA) that was used to collect data at 100 Hz with custom-built software; motorized high-precision positioning stages for x-direction (M-404.8PD, 0.50 μm precision, Physik Instrumente GmbH, Germany), y-direction (M-122.2DD, 0.15 μm precision, Physik Instrumente GmbH, Germany), and z-

direction (M-111.1DG, 0.10 μm precision, Physik Instrumente GmbH, Germany); a piezoelectric microdispenser (PicPIP, 100 pl minimum droplet size, GeSiM GmbH, Germany); a high-speed camera sideview (Phantom Miro LC310, Vision Research Inc., USA) with a 1–5× macro lens (MP-E 65, Canon Inc., Japan) and another CCD camera sideview (IGV B1320M, Imperx Inc., USA) with a 1–6× microscope lens (VZM 600i, Edmund Optics Inc., USA). Measurements were carried out in room temperature (~22 °C) and in ~40% relative humidity.

**Sensor probe preparation**. 1 mm diameter SU-8 discs were used for holding the probe droplet. The SU-8 discs were fabricated by UV lithography. First, 150 nm of aluminium was sputtered on top of a silicon wafer. The wafer was then baked overnight in an oven at 120 °C. Next, a 80 μm thick layer of SU-8 50 was applied on the wafer by spin coating (1500 rpm, 30 s), baked for 15 min at 95 °C exposed for 20 s (Karl Suss MA6 mask aligner) and baked for 15 min at 95 °C. The probes were then released by heating the wafer to 200 °C and cooling to room temperature, which causes thermal expansion mismatch stress that releases the SU-8 from the aluminium. The SU-8 discs were then sputtered with gold on both sides and glued at the end of the force sensor tip using UV-curable glue (Byllux 5118, Byla GmbH, Germany). A droplet is deposited on the SU-8 disc by shooting sub-nl droplets using the piezoelectric microdispenser. The small droplets bounce off a super-hydrophobic surface onto the disc, accumulating into a larger droplet used in the measurements. The volume of the droplet is measured through the force given by the sensor.

**Procedure for measuring individual points**. A single measurement starts with the sample surface brought close to a water droplet pinned on the sensor probe. The sample stage starts moving up towards the droplet at a constant speed (2 μm s$^{-1}$ for accuracy and repeatability experiments, 5 μm s$^{-1}$ for all other experiments) until the droplet touches the surface (snap-in), i.e., a step down is detected in the force. The stage keeps moving up either for a fixed time after snap-in, or until the force reaches a predetermined value. For any set of experiments, either the fixed time delay or the fixed force limit was used to minimize variation between measurements. The maximum force depends on the distance moved up during the time delay, or is directly the preset force limit. The stage is then retracted down at 10 μm s$^{-1}$ until the droplet separates from the surface (pull-off), i.e., a step up is detected in the force. The volume of water droplet used in the measurements is 1.5 μl. After each measurement, the droplet is refilled to 1.5 μl using the microdispenser. The above-mentioned low approach speeds (2 μm s$^{-1}$ or 5 μm s$^{-1}$), and moderate retraction speed (10 μm s$^{-1}$) were chosen to avoid dynamic effects while keeping the measurement duration reasonable.

**Approach/retraction speed characterization**. The effect of approach speed on snap-in force, and the effect of retraction speed on pull-off force, was characterized for five different speeds. Measurements were carried out on a fluoropolymer-coated silicon nanograss surface (slightly less superhydrophobic than the one listed in Supplementary Table 2). A force vs. speed comparison is shown in Supplementary Fig. 8.

**Automated scanning**. The measurements were fully automated for accuracy and repeatability testing on micropillars, line scanning, and area scanning of wetting maps. The force readout of the probe is used as feedback to detect contact when probing a surface, and to monitor the droplet volume when refilling the droplet after every measurement. Measurement locations are generated beforehand, but after that hundreds of measurements can be carried out automatically without human intervention.

**Data analysis**. Snap-in and pull-off events are extracted from individual force recordings using a step detection algorithm based on Student's t-test metric:

$$t = \frac{\overline{x}_1 - \overline{x}_2}{\sqrt{\frac{s_1^2}{N} + \frac{s_2^2}{N}}}, \tag{1}$$

where, for each force value in the signal, $\overline{x}_1$ is the mean of the $N$ previous values, $\overline{x}_2$ is the mean of the $N$ subsequent values, $s_1^2$ is the variance of the $N$ previous values, and $s_2^2$ is the variance of the $N$ subsequent values. The threshold $t$ is adjusted based on the magnitude of measured forces and the sensor noise. In practice $t$ varied from 20 to 80. The value of $N$ depends on the force data sampling rate and was set to 8 for the 100 Hz used in the measurements. After detection of snap-in event in the force recording (time of snap-in), polynomial curves are fitted to the preceding and succeeding values, so that magnitude of the step (snap-in force) can be extracted from the difference between the fitted curves at the time of the event. Pull-off force is determined by the difference in the force after the detected pull-off event (time of pull-off) and the minimum force in the recording.

**Numerical simulation**. For detailed discussion on modelling and simulation, see Supplementary Note 1.

**Contact angle measurements**. Contact angles were measured using sessile drop method by Attension Theta optical tensiometer with automated liquid pumping system. Advancing contact angles were measured by placing a 0.2 µl droplet on the surface and increasing its volume to 40 µl, at a rate of 0.10 µl s⁻¹. Receding contact angles were measured by decreasing droplet volume at a rate of 0.10 µl s⁻¹, starting with a drop volume of 40 µl.

**Data availability**. The data that support the findings of this study are available from the corresponding author on reasonable request.

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

## Acknowledgements

This work was supported by the European Research Council ERC-2016-CoG (725513-SuperRepel), Academy of Finland (Centres of Excellence Programme (2014-2019) and projects #295006, #296250, #266820, #256206, #263560, #268685, #292477, #297360, and #299087), the AScI/ELEC Thematic Research Programme, Aalto ELEC Doctoral School, Finnish Society of Automation, and Finnish Cultural Foundation. This work made use of the Aalto University Nanomicroscopy Centre premises and the cleanroom facilities of Micronova, Centre for Micro and Nanotechnology.

## Author contributions

R.H.A.R. conceived the research. Q.Z. designed the apparatus. Q.Z., R.H.A.R., V.L. and M.V. designed the sensor probe. V.L. and M.V. constructed the apparatus. V.L. and Q.Z. designed the measurement algorithms. R.H.A.R., Q.Z., M.V., V.L. and V.J. designed the experiments and analysed the results. M.V., V.L. and M.J.H. performed experiments. V.J. fabricated the sensor probe disc and microstructured Si samples. V.S. carried out modelling and numerical simulations. All authors discussed and co-wrote the paper.

## Additional information

**Competing interests:** The authors have filed a patent application based on the content of the manuscript.

