## [Peer Review File · Nature Communications]

Reviewers' comments:

Reviewer #1 (Remarks to the Author):

This is an elegant piece of work. The authors developed a wetting based method to explore the heterogeneity of a surface, which is likely to have important consequences in the characterization of real surfaces. There is a minor issue regarding heterogeneity versus the ensuing instability, which is partly semantic, partly philosophical. As the microscopic contact line gets pinned and depinned, it gives out a signal that can manifest in different ways. One is in the force fluctuation, which is the basis of this study. The noise that blooms out during this process can also excite the droplet and lead to various types dynamics. Such a phenomenon was reported in *Eur. Phys. J. E* (2013) 36: 15 and reviewed in *Langmuir*, 2015, 31 (34), pp 9266–9281. The method described by the authors would nicely complement the works reported in the above two papers. In that sense, I would have thought that citations of those papers might be more relevant to this paper than many of the other cited papers. In any event, the authors need to be careful in what the measurements mean beyond a certain point as an important length scale, the Larkin length, would play a role in the way a contact line detaches that might rectify the response received from force fluctuation.

Reviewer #3 (Remarks to the Author):

The paper entitled "Mapping microscale wetting variations on biological and synthetic water-repellent surfaces" reports a technique of scanning droplet adhesion microscopy to create wetting maps with a sensing resolution in pico newton range and a spatial resolution of 10 microns. The wetting force (snap-in force) and dewetting force (pull-off force) have been obtained on various hydrophobic and superhydrophobic surfaces, exemplified by butterfly wings and synthesized surfaces with microscale spatial heterogeneity. The results show that droplet adhesion correlates well with water-repellency and is a sensitive measure for superhydrophobic surface characterization.

This is a nice paper. But I think the authors need more in-depth discussion on their results.

(1) In Fig. 1b, it seems that the adhesion force is shown as a negative force, indicated by a sudden decrease in the force plot for a snap-in force, and negative forces during the depinning steps.

What does the maximum force represent? On the super-repellent surfaces, it is exactly the repellent force, shown as a positive force. But it depends on the distance between the probing droplet and the sample surface. More specifically, on the hydrophobic surfaces in Fig. 4e and f in the extended data, there is no repellent force. During the entire measurement, all the interaction forces are negative, i.e., adhesion forces, no repellent forces. This could be an additional metric to define repellent surfaces.

(2) In Fig. 2, the simulation results need more discussion, why discrepancy in snap-in forces occurs for smaller radius pillars, etc.

(3) In Fig. 3, it seems that each of the snap-in event (P1-P16) should correspond to contact line advancing to each pillar. And the depinning events should be the same number as pinning events. Well, some of the depinning could be combined and become indiscernible. D7 and D8 could have some penetration into the pillars. The difference between the initial and final force also indicates significant weight loss of 6–7 μN . The discussion in the example in Fig. 3 should be more in-depth to elaborate the wetting and dewetting dynamics, which showcases that the scanning droplet adhesion microscopy can be employed to study the microscale interfacial liquid-solid interactions.

(4) The authors need to comment on the force difference between the initial and final forces. The force difference or weight loss (shown as an increase in the force plot) are due to evaporation and the droplet offset to the substrate, during droplet retraction or pull-off (extended data in Fig. 4f)

(5) In the procedure for measuring individual points, the authors mentioned "the sample stage starts moving up... until the droplet touches the surface (snap-in)". How to define "touch"? Also, "after a short time in contact the stage is then retracted...", does the stage continue moving up after "touching" the surface or stay still in contact? Please be more precise. The maximum force on the force plot is probably a function of the moving up distance.

Reviewer #4 (Remarks to the Author):

This paper reports on a droplet-mapping method to explore the wetting heterogeneity of structured surfaces. This is achieved by attaching a small drop to a force sensor and repeatedly contacting it with the surface under interrogation in a spatially resolved manner. The obtained data for various microstructured surfaces is presented as adhesion and snap-in forces, and there are some efforts to compare with numerical simulations.

Overall I think the paper is rather interesting, and presents the obtained data well. The technique appears to be quite useful. However, in this reviewer's opinion, the impact and substance of the present work is somewhat limited by the systems chosen and the lack of contextualisation to well known and well characterised wetting systems. It seems that obvious opportunities to explain better the surface thermodynamics are missed, and this would need to be tackled before publication in a journal of this type. At the moment, the paper essentially reads as a methods article, albeit a very neat and well presented one. The new science is not evident in the current format.

I have some suggestions for large items to tackle in revision below (along with some more minor gripes) that should be addressed before publication in order to make the paper more useful to other practitioners in the field.

Major issues:

- 1) The most glaring omission to my eye is that there is no comparison between measurements made using this new system and typical wetting measurements on non-structured (flat) surfaces, where the macroscopic contact angle is well-defined. This would make more a much more compelling proof of concept, and would allow for a more scientific approach to linking the thermodynamics of wetting to this new measurement approach. This leads onto point 2:
- 2) The comparison between expected wetting characteristics and measurements is vague and unsatisfying. The surface evolver simulations are not shown in detail anywhere, and we are essentially expected to trust on sight the data in Figs 2 d,e. In any case, surface evolver is not ideal for the purpose, and a simple calculation of the depinning/energy change on separation of the droplet and surface would be much more compelling. Overall, the lack of context for these measurements is an issue.
- 3) Leading on from the issues above, the measurement of a droplet adhesion force is fundamentally different to measuring a microscopic or indeed macroscopic contact angle. There is one vague statement early on that the two are linked (which is not untrue, but the relationship is far from simple), and so the authors should be careful about claims that they make, and precisely what their measurements are showing.

Minor issues/queries:

- 4) Structures surfaces can tend to be very fragile, and contaminate easily – does the droplet become contaminated? Extended data fig 2 appears to show a systematic decrease in the measured forces. Is this consistent with contamination of the droplet?
- 5) Does not allow back-conversion to contact angle, so have measured adhesion properties, not microscopically mapped contact angle.
- 6) Was the interaction speed used optimised? Can we see a plot of the effective forces as a function of speed?
- 7) I'm not a personal fan of the colour/contouring method used in e.g. Fig 1 c,d, as it is misleading as to the density and spatial fidelity of the data. I prefer a simple 'pixel' for each measurement coloured with an appropriate solid fill.

Response to Referees for NCOMMS-17-10728-T

We thank the reviewers for their helpful suggestions and constructive comments. We have addressed all the concerns raised by the reviewers. As a result, the paper has been substantially revised, including additional data and modeling, extended discussion, clarification of issues, one new Extended Data Figure, and one updated figure in the main text. The major changes are:

1. We have added an extensive Supplementary Note (10 pages) that explains in detail the modeling and simulation of the droplet adhesion forces, including excerpts of code for running the simulation in Matlab/Simulink. Short discussion on the simulation has also been added to the main text.
2. Simulation results on micropillars presented in Fig. 2d,e have been updated, where the earlier discrepancy between simulation and experimental results on the smallest pillars has been resolved.

Changes to the text are highlighted in the revised manuscript file in yellow. In the following, we address the reviewers' comments one by one.

Reviewer #1:

General comment

This is an elegant piece of work. the authors developed a wetting based method to explore the heterogeneity of a surface, which is likely to have important consequences in the characterization of real surfaces.

Our response

We thank the reviewer for appreciating our approach to characterizing heterogeneity of surfaces, and for acknowledging the potential of our work.

Comment 1

There is a minor issue regarding heterogeneity versus the ensuing instability, which is partly semantic, partly philosophical. As the microscopic contact line gets pinned and depinned, it gives out a signal that can manifest in different ways. One is in the force fluctuation, which is the basis of this study. The noise that blooms out during this process can also excite the droplet and lead to various types dynamics. Such a phenomenon was reported in Eur. Phys. J. E (2013) 36: 15 and reviewed in Langmuir, 2015, 31 (34), pp 9266–9281. The method described by the authors would nicely complement the works reported in the above two papers. In that sense, I would have thought that citations of those papers might be more relevant to this paper than many of the other cited papers.

Our response

Thank you for the comment. We have added a sentence to the main text, including citations to the mentioned studies of droplet oscillations due to pinning and depinning on pillar surfaces (lines 123 – 124):

“Droplets rolling off such micropillar surfaces may oscillate due to the pinning and depinning steps^{31,32},”

Comment 2

In any event, the authors need to be careful in what the measurements mean beyond a certain point as an important length scale, the Larkin length, would play a role in the way a contact line detaches that might rectify the response received from force fluctuation.

Our response

Thank you for the comment, this is a very interesting and valid point. We understand that for the extremely repellent surfaces the contact radius might indeed be smaller than the length scale of the pinning points. The current work is only the first step towards developing a method to identify the local and spatially varying relation between micro- and nanoscale surface features and macroscale wetting behavior. In the future we will systematically study the importance of parameters such as the Larkin length.

Reviewer #3:

General comment

The paper entitled “Mapping microscale wetting variations on biological and synthetic water-repellent surfaces” reports a technique of scanning droplet adhesion microscopy to create wetting maps with a sensing resolution in pico newton range and a spatial resolution of 10 microns. The wetting force (snap-in force) and dewetting force (pull-off force) have been obtained on various hydrophobic and superhydrophobic surfaces, exemplified by butterfly wings and synthesized surfaces with microscale spatial heterogeneity. The results show that droplet adhesion correlates well with water-repellency and is a sensitive measure for superhydrophobic surface characterization.

This is a nice paper. But I think the authors need more in-depth discussion on their results.

Our response

We thank the reviewer for all the comments. We agree that more discussion is needed and have addressed the issue in many places.

Comment 1

In Fig. 1b, it seems that the adhesion force is shown as a negative force, indicated by a sudden decrease in the force plot for a snap-in force, and negative forces during the depinning steps. What does the maximum force represent? On the super-repellent surfaces, it is exactly the repellent force, shown as a positive force. But it depends on the distance between the probing droplet and the sample surface. More specifically, on the hydrophobic surfaces in Fig. 4e and f in the extended data, there is no repellent force. During the entire measurement, all the interaction forces are negative, i.e., adhesion forces, no repellent forces. This could be an additional metric to define repellent surfaces.

Our response

Thank you for the comment, we should clarify directions of the forces. Both snap-in and pull-off are attractive forces as they represent adhesion between the droplet and the surface. The direction of the forces on the force curve (negative/positive) depends on whether the force level after the step is lower or higher so that snap-in = negative step and pull-off = positive step. This has been clarified under “Procedure for measuring individual points” in the Methods section (lines 362 – 370):

“The sample stage starts moving up towards the droplet at a constant speed (2 $\mu\text{m/s}$ for accuracy and repeatability experiments, 5 $\mu\text{m/s}$ for all other experiments) until the droplet touches the surface (snap-in), i.e. a step down is detected in the force. The stage keeps moving up either for a fixed time after snap-in, or until the force reaches a predetermined value. For any set of experiments, either the fixed time delay or the fixed force limit was used to minimize variation between measurements. The maximum force depends on the distance moved up during the time delay, or is directly the preset force limit. The stage is then retracted down at 10 $\mu\text{m/s}$ until the droplet separates from the surface (pull-off), i.e. a step up is detected in the force.”

The maximum force i.e. the highest value on the force curve is usually found at the point where the stage starts retracting back down, and depends on how far the sample was pushed up after snap-in.

As the reviewer pointed out, this is not the case for hydrophobic surfaces (Extended Data Fig. 4f), where the force exerted by the upwards moving sample stage through the meniscus is countered by wetting to the sides i.e. rapidly increasing contact area and thus increasing downward surface tension force. Therefore, the maximum force remains well below the level before snap-in. However, the maximum force still depends on the measurement procedure (how much the sample stage is driven up). Therefore, we prefer using snap-in and pull-off, which for a given droplet size are independent of the stage movements.

Comment 2

In Fig. 2, the simulation results need more discussion, why discrepancy in snap-in forces occurs for smaller radius pillars, etc.

Our response

We agree that the discussion on simulation was lacking. To rectify the issue, an extensive Supplementary Note on modeling and simulation has been added. We also discovered the reason for

the discrepancy of snap-in forces and simulation on the smallest pillars. The simulation results presented in Fig. 2 did not account for the fact that the droplet pins on the lower edge of the pillar cap³³, because the cap is made of highly wettable silicon dioxide. Consequently, there is a vertical offset, equal to the thickness of the pillar cap (1.2 μm), in the distance between the base of the droplet and the pillar it is touching. When this offset is taken into account, the snap-in force simulation matches well with experimental data for all pillar radii.

Figure 2d,e have been updated with simulation results where the thickness of the pillar cap is taken into account. Short discussion has also been added to the main text (lines 100 – 103):

“The simulation uses Young-Laplace equation and boundary conditions to solve the shape of the droplet, which can be used to compute the forces¹⁸. A detailed discussion on modelling and simulation can be found in Supplementary Note 1.”

Due to addition of the Supplementary Note that fully covers modeling and simulation, the short paragraph “Numerical simulation” in the Methods section has been replaced with a short reference to Supplementary Note 1 (lines 397 – 398):

“For detailed discussion on modelling and simulation, see Supplementary Note 1.”

Comment 3

In Fig. 3, it seems that each of the snap-in event (P1-P16) should correspond to contact line advancing to each pillar. And the depinning events should be the same number as pinning events. Well, some of the depinning could be combined and become indiscernible. D7 and D8 could have some penetration into the pillars. The difference between the initial and final force also indicates significant weight loss of 6~7 μN . The discussion in the example in Fig. 3 should be more in-depth to elaborate the wetting and dewetting dynamics, which showcases that the scanning droplet adhesion microscopy can be employed to study the microscale interfacial liquid-solid interactions.

Our response

We agree that commenting on the wetting and dewetting should be added. The number of pinning and depinning steps does not match, because some of the depinning steps are indeed combined into larger steps when the droplet depins from multiple pillars simultaneously. Discussion has been added to the main text, along with acknowledgment that the volume loss during this particularly long measurement (close to 8 minutes vs about half a minute for a typical standard measurement) may affect the magnitude of the pinning/depinning steps (lines 128 – 137):

“The number of pinning steps in Fig. 3 is much higher than the number of depinning steps, because depinning from multiple pillars are combined into larger steps. It should be noted that the magnitude of pinning steps shown in Fig. 3c corresponds to snap-in force on an individual pillar, shown in Fig. 1f. On the other hand, the dynamics of collective depinning involves lateral deformation of the meniscus, so the magnitude of depinning steps in normal direction (Fig. 3c) is smaller than pull-off on individual pillars (Fig. 1f), except for the final depinning step. It should also be noted that during

a long measurement such as the one presented in Fig. 3, the volume loss due to evaporation of the droplet may affect the magnitude of the pinning and depinning steps (see Supplementary Note 1 for further theoretical discussion on the volume dependency).”

Comment 4

The authors need to comment on the force difference between the initial and final forces. The force difference or weight loss (shown as an increase in the force plot) are due to evaporation and the droplet offset to the substrate, during droplet retraction or pull-off (extended data in Fig. 4f)

Our response

Weight loss by evaporation is inevitable during the experiment and should be addressed. The volume dependency of the snap-in and pull-off forces is theoretically discussed in Supplementary Note 1. At the end of the description of the setup (lines 75 – 77 in the main text) we conclude that evaporation is not significant during a typical measurement.

However, the reviewer correctly pointed out a significant weight loss visible in the force curve of Extended Data Fig. 4f. The surface in question is a flat silicon wafer coated with a fluoropolymer, and it was in fact the only surface we measured that left a droplet behind at pull-off. This should have been noted in the caption, which we have now modified (lines 466 – 469) with the following addition:

“Note the volume loss visible in panel f, where the final force level is significantly higher than the initial value. The flat surface has a macroscopically measured receding contact angle of less than 90° (see Extended Data Table 2) and leaves a droplet behind on the surface at pull-off.”

Comment 5

In the procedure for measuring individual points, the authors mentioned “the sample stage starts moving up... until the droplet touches the surface (snap-in)”. How to define “touch”? Also, “after a short time in contact the stage is then retracted...”, does the stage continue moving up after “touching” the surface or stay still in contact? Please be more precise. The maximum force on the force plot is probably a function of the moving up distance.

Our response

These are all important points, the relevant terms in the measurement procedure must be clearly defined. We define "touching" as detecting the snap-in event i.e. a negative step in the force (using the same Student's t-test as was used in offline data analysis afterwards, see “Data analysis” in the Methods section). The stage continues to move up for a short while after snap-in, either for a fixed time or until a predetermined force value is reached. The purpose is to minimize variation between measurements. We have clarified the “Procedure for measuring individual points” in the Methods section and commented on the maximum force (lines 362 – 370):

“The sample stage starts moving up towards the droplet at a constant speed (2 $\mu\text{m/s}$ for accuracy and repeatability experiments, 5 $\mu\text{m/s}$ for all other experiments) until the droplet touches the surface (snap-in), i.e. a step down is detected in the force. The stage keeps moving up either for a fixed time after snap-in, or until the force reaches a predetermined value. For any set of experiments, either the fixed time delay or the fixed force limit was used to minimize variation between measurements. The maximum force depends on the distance moved up during the time delay, or is directly the preset force limit. The stage is then retracted down at 10 $\mu\text{m/s}$ until the droplet separates from the surface (pull-off), i.e. a step up is detected in the force.”

Reviewer #4:

General comment

This paper reports on a droplet-mapping method to explore the wetting heterogeneity of structured surfaces. This is achieved by attaching a small drop to a force sensor and repeatedly contacting it with the surface under interrogation in a spatially resolved manner. The obtained data for various microstructured surfaces is presented as adhesion and snap-in forces, and there are some efforts to compare with numerical simulations.

Overall I think the paper is rather interesting, and presents the obtained data well. The technique appears to be quite useful. However, in this reviewer’s opinion, the impact and substance of the present work is somewhat limited by the systems chosen and the lack of contextualisation to well known and well characterised wetting systems. It seems that obvious opportunities to explain better the surface thermodynamics are missed, and this would need to be tackled before publication in a journal of this type. At the moment, the paper essentially reads as a methods article, albeit a very neat and well presented one. The new science is not evident in the current format.

I have some suggestions for large items to tackle in revision below (along with some more minor gripes) that should be addressed before publication in order to make the paper more useful to other practitioners in the field.

Our response

We are grateful for all the constructive comments, and have made substantial efforts to follow the suggestions.

Major issues:

Comments 1 & 2

The most glaring omission to my eye is that there is no comparison between measurements made using this new system and typical wetting measurements on non-structured (flat) surfaces, where the

macroscopic contact angle is well-defined. This would make more a much more compelling proof of concept, and would allow for a more scientific approach to linking the thermodynamics of wetting to this new measurement approach. This leads onto point 2: The comparison between expected wetting characteristics and measurements is vague and unsatisfying. The surface evolver simulations are not shown in detail anywhere, and we are essentially expected to trust on sight the data in Figs 2 d,e. In any case, surface evolver is not ideal for the purpose, and a simple calculation of the depinning/energy change on separation of the droplet and surface would be much more compelling. Overall, the lack of context for these measurements is an issue.

Our response

Thank you for the comment. We agree that the discussion on the numerical simulation and the context overall was lacking. We have added an extensive Supplementary Note on modeling and simulation. Instead of Surface Evolver, the new numerical simulation method is based on solving the Young-Laplace equation as a boundary value problem. Both methods give the same simulation results within numerical accuracy. The modelled relationship between the snap-in and pull-off forces and work of adhesion is also discussed in the Supplementary Note (see Figure N9). Additionally, Extended Data Table 2 shows measured snap-in and pull-off forces for various superhydrophobic (structured, rough) surfaces with measured contact angles, and two hydrophobic (non-structured, flat) surfaces. Our work is focused on superhydrophobic surfaces that are challenging to measure with other techniques. A systematic study (see ref. 19) on droplet adhesion forces on non-structured, flat surfaces has been previously conducted using an instrument three orders of magnitude less sensitive than ours. In that study, the relationship between the forces and contact angles on flat surfaces was analysed. We have added a short discussion and a reference to the Supplementary Note to the main text (lines 100 – 103):

“The simulation uses Young-Laplace equation and boundary conditions to solve the shape of the droplet, which can be used to compute the forces¹⁸. A detailed discussion on modelling and simulation can be found in Supplementary Note 1.”

We also discovered the reason for the discrepancy of snap-in forces and simulation on the smallest pillars. The simulation results presented in Fig. 2 did not account for the fact that the droplet pins on the lower edge of the pillar cap, because the cap is hydrophilic silicon dioxide. Consequently, there is a vertical offset, equal to the thickness of the oxide cap (1.2 μ m), in the distance between the base of the droplet and the pillar it is touching. When this offset is taken into account, the snap-in force simulation matches well with experimental data for all pillar radii.

Figure 2d,e have been updated with simulation results where the thickness of the pillar cap is taken into account. Due to addition of the Supplementary Note that fully covers modeling and simulation, the short paragraph “Numerical simulation” in the Methods section has been replaced with a short reference to Supplementary Note 1 (lines 397 – 398):

“For detailed discussion on modelling and simulation, see Supplementary Note 1.”

Comment 3

Leading on from the issues above, the measurement of a droplet adhesion force is fundamentally different to measuring a microscopic or indeed macroscopic contact angle. There is one vague statement early on that the two are linked (which is not untrue, but the relationship is far from simple), and so the authors should be careful about claims that they make, and precisely what their measurements are showing.

Our response

Thank you for the important comment, we should definitely be precise about our claims. Measuring droplet adhesion is certainly different from measuring contact angles. The sentence mentioning the connection between the measured forces and contact angles cites the work of Samuel et al., where a systematic experimental study of the relationship is presented. We have clarified the sentence in the main text (lines 50 – 53):

“It has been experimentally verified that snap-in (first droplet contact) and pull-off (droplet separation) adhesion forces on hydrophobic surfaces are related to respectively the advancing contact angle (θ_{adv}) and the receding contact angle (θ_{rec})¹⁹. Smaller forces correspond to larger contact angles.”

Furthermore, the newly added Supplementary Note describes the model that gives qualitative indication of the link between the measured forces and contact angles, and shows close to linear relationship between the forces and work of adhesion (see Figure N9 in Supplementary Note 1).

We certainly do not want to claim that we are probing microscopic or macroscopic contact angles, rather we are probing a microscopic, extremely localized property of the surface (droplet adhesion) that is connected to wettability. To clarify this, we have extended one of the key sentences in the introduction (lines 25 – 27) to be more precise:

“Here we introduce wetting maps as a new concept for visualizing imperfections and local variations in wetting through droplet adhesion forces, which correlate with wettability.”

Minor issues/queries:

Comment 4

Structures surfaces can tend to be very fragile, and contaminate easily – does the droplet become contaminated? Extended data fig 2 appears to show a systematic decrease in the measured forces. Is this consistent with contamination of the droplet?

Our response

Thank you for the comment, this is a valid concern. We planned the experiments in a way that minimizes effects both from the environment and the surface itself. The accuracy and repeatability measurements on the micropillars were carried out by repeating a pillar-to-pillar scan ten times (see the sketch below), which minimizes time difference between measurement on the first pillar and measurement on the last pillar. Consequently, time dependent effects, such as potential

contamination or changes in ambient conditions, are minimized *between the pillars* and appear (if any) *within the pillars*. The ANOVA analysis of the results (Extended Data Table 1) shows that the differences between the pillars are statistically significant, compared to variations within individual pillars (which may include said time dependent effects).

Measurement procedure for accuracy and repeatability measurements on micropillars.

Comment 5

Does not allow back-conversion to contact angle, so have measured adhesion properties, not microscopically mapped contact angle.

Our response

The model presented in Supplementary Note 1 allows conversion from measured forces to contact angles for ideal surfaces. We have also clarified a key sentence in the introduction to state more precisely what we are measuring (lines 25 – 27):

“Here we introduce wetting maps as a new concept for visualizing imperfections and local variations in wetting through droplet adhesion forces, which correlate with wettability.”

Comment 6

Was the interaction speed used optimised? Can we see a plot of the effective forces as a function of speed?

Our response

Thank you for the constructive comment. We chose very low approach speeds (2 $\mu\text{m/s}$ for the repeatability and sensitivity measurements, 5 $\mu\text{m/s}$ for all other measurements) and moderate retraction speeds (10 $\mu\text{m/s}$ for all measurements) to avoid dynamic effects while keeping the

measurement duration reasonable. We have added the force vs speed plot to the new Extended Data Fig. 8 and a short discussion to “Procedure for measuring individual points” in the Methods section (lines 371 – 373):

“The above mentioned low approach speeds (2 $\mu\text{m/s}$ or 5 $\mu\text{m/s}$), and moderate retraction speed (10 $\mu\text{m/s}$) were chosen to avoid dynamic effects while keeping the measurement duration reasonable.”

A new section, “Approach/retraction speed characterization”, has been added to the Methods section (lines 374 – 378):

“Approach/retraction speed characterization. The effect of approach speed on snap-in force, and the effect of retraction speed on pull-off force, was characterized for five different speeds. Measurements were carried out on a fluoropolymer coated silicon nanoglass surface (slightly less superhydrophobic than the one listed in Extended Data Table 2). A force vs speed comparison is shown in Extended Data Fig. 8.”

Comment 7

I'm not a personal fan of the colour/contouring method used in e.g. Fig 1 c,d, as it is misleading as to the density and spatial fidelity of the data. I prefer a simple 'pixel' for each measurement coloured with an appropriate solid fill.

Our response

We understand the concern, which is why we intentionally had added black dots to represent measurement points, helping the reader interpreting density and spatial fidelity of the data. We have tested many different ways of plotting the maps, and have chosen the contour map as we feel it works well for data visualization in this case. However, the corresponding bitmaps are provided below.

REVIEWERS' COMMENTS:

Reviewer #3 (Remarks to the Author):

The authors have thoroughly addressed the points I raised. The revised manuscript looks very good to me.

One minor point: in the Supplementary Note, the title and authors may need to be included.

Reviewer #4 (Remarks to the Author):

Having read carefully the revised manuscript and the authors' response to the reviewer comments, I am pleased to say that I see the manuscript is ready for publication in Nature Communications. The impact is substantially increased by a more thorough and robust analysis of the approach and obtained data.

The authors have worked hard to address the major concerns of the reviewers, and in doing so, have much improved aspects of the paper. From my particular perspective, the modelling is now very satisfactory and makes a more substantial contribution to the outcome of the work, as well as holding significant pedagogic value for readers. Well done!

Although I still prefer the simple bitmaps for adhesion data that the authors provide in response to my last query (they are so much prettier!) I'll happily concede this point in order for this nice manuscript to now be accepted for publication.

My recommendation is therefore to accept the paper without further revision.